# Ventilatory Effects of Isoflurane Sedation via the Sedaconda ACD-S versus ACD-L: A Substudy of a Randomized Trial

**DOI:** 10.3390/jcm12093314

**Published:** 2023-05-06

**Authors:** Lukas M. Müller-Wirtz, Tobias Becher, Ulf Günther, Martin Bellgardt, Peter Sackey, Thomas Volk, Andreas Meiser

**Affiliations:** 1Department of Anaesthesiology, Intensive Care and Pain Therapy, Saarland University Medical Center and Saarland University Faculty of Medicine, 66421 Homburg, Germany; thomas.volk@uks.eu; 2Outcomes Research Consortium, Cleveland, OH 44195, USA; 3Department of Anesthesiology and Intensive Care Medicine, Campus Kiel, University Medical Center Schleswig-Holstein, 24118 Kiel, Germany; tobias.becher@uksh.de; 4Department of Anaesthesiology, Intensive Care, Emergency Medicine, Pain Therapy, University Hospital Oldenburg, 26133 Oldenburg, Germany; guenther.ulf@klinikum-oldenburg.de; 5Department of Anaesthesiology and Intensive Care Medicine, St. Josef-Hospital, University Hospital of the Ruhr-University Bochum, 44780 Bochum, Germany; martin.bellgardt@kklbo.de; 6Unit of Anesthesiology and Intensive Care, Department of Physiology and Pharmacology, Karolinska Institute, 17177 Stockholm, Sweden; peter.sackey@ki.se; 7Sedana Medical AB, 18232 Danderyd, Sweden

**Keywords:** anesthesia, critical care, intensive care, sedation, volatile anesthetic, isoflurane, propofol

## Abstract

Devices used to deliver inhaled sedation increase dead space ventilation. We therefore compared ventilatory effects among isoflurane sedation via the Sedaconda ACD-S (internal volume: 50 mL), isoflurane sedation via the Sedaconda ACD-L (100 mL), and propofol sedation with standard mechanical ventilation with heat and moisture exchangers (HME). This is a substudy of a randomized trial that compared inhaled isoflurane sedation via the ACD-S or ACD-L to intravenous propofol sedation in 301 intensive care patients. Data from the first 24 h after study inclusion were analyzed using linear mixed models. Primary outcome was minute ventilation. Secondary outcomes were tidal volume, respiratory rate, arterial carbon dioxide pressure, and isoflurane consumption. In total, 151 patients were randomized to propofol and 150 to isoflurane sedation; 64 patients received isoflurane via the ACD-S and 86 patients via the ACD-L. While use of the ACD-L was associated with higher minute ventilation (average difference (95% confidence interval): 1.3 (0.7, 1.8) L/min, *p* < 0.001), higher tidal volumes (44 (16, 72) mL, *p* = 0.002), higher respiratory rates (1.2 (0.1, 2.2) breaths/min, *p* = 0.025), and higher arterial carbon dioxide pressures (3.4 (1.2, 5.6) mmHg, *p* = 0.002), use of the ACD-S did not significantly affect ventilation compared to standard mechanical ventilation and sedation. Isoflurane consumption was slightly less with the ACD-L compared to the ACD-S (−0.7 (−1.3, 0.1) mL/h, *p* = 0.022). The Sedaconda ACD-S compared to the ACD-L is associated with reduced minute ventilation and does not significantly affect ventilation compared to a standard mechanical ventilation and sedation setting. The smaller ACD-S is therefore the device of choice to minimize impact on ventilation, especially in patients with a limited ability to compensate (e.g., COPD patients). Volatile anesthetic consumption is slightly higher with the ACD-S compared to the ACD-L.

## 1. Introduction

While sedation remains an important component of intensive care, most commonly used intravenous sedatives may accumulate or cause substantial harm during prolonged periods of use [1,2,3,4]. Therefore, volatile anesthetics are increasingly used as an alternative for sedation in intensive care [5,6,7,8].

Volatile anesthetic reflection devices were designed to enable inhaled sedation with open circuit ventilators which are commonly used in intensive care [9,10]. A potential drawback is that these devices may increase dead space ventilation, leading to compensatory increases in minute ventilation to maintain carbon dioxide elimination. Thus, it is unclear if low tidal volume ventilation during inhaled sedation can be maintained [11]. A new version of the Anaesthetic Conserving Device (Sedaconda ACD; Sedana Medical AB, Danderyd, Sweden) with an internal volume of 50 mL (ACD-S) compared to 100 mL of the original version (ACD-L) was therefore designed. Experimental and initial small clinical studies suggest a reduction in minute ventilation and comparable efficiency of volatile anesthetic reflection with the ACD-S compared to the ACD-L [12,13,14]. However, larger investigations are still needed to evaluate the dead space effect of anesthetic conserving devices under clinical conditions in common intensive care settings.

This substudy of a randomized controlled trial primarily aimed to assess the ventilatory effects of dead space reduction in a volatile anesthetic reflection device used for isoflurane sedation in invasively ventilated intensive care patients. Specifically, we hypothesized that the use of the ACD-S reduces minute ventilation compared to the ACD-L in intensive care patients over the first 24 h of sedation. Additionally, we compared the use of ACD-S or ACD-L to a standard mechanical ventilation with heat and moisture exchangers (HMEs) under propofol sedation. Secondarily, we compared the volatile anesthetic consumption between the ACD-S and ACD-L under clinical conditions.

## 2. Materials and Methods

The underlying multicentric randomized controlled trial was registered in the European Medicines Agency’s EU Clinical Trial register (EudraCT, https://www.clinicaltrialsregister.eu/, ID: 2016–004551–67), before including the first patient. The presented substudy was approved by the responsible Institutional Review Board (IRB; approval number: 11/17, Ethikkommission der Ärztekammer des Saarlandes, Saarbrücken, Germany) and retrospectively registered at the German Clinical Trials Register (DRKS, https://www.drks.de/, ID: DRKS00020240) on 7 January 2020, before opening the underlying trial’s database and obtaining access to study results. Written informed consent was obtained from all subjects or a legal surrogate for participation in the underlying trial, and additional consent for the presented substudy was waived by the IRB. This manuscript adheres to the applicable STROBE guidelines.

### 2.1. Study Design

This is a prospective substudy of a multicentric randomized controlled trial with the primary objective of investigating the influence of the volatile anesthetic conserving devices Sedaconda ACD-S and ACD-L (Sedana Medical AB, Danderyd, Sweden) on ventilation parameters.

The underlying trial investigated the non-inferiority of isoflurane to propofol sedation in intensive care patients, and included patients from 13 medical or general, 10 surgical, and 1 neurological intensive care units in Germany (21 sites) and Slovenia (3 sites) [7]. Adult medical and surgical intensive care patients with mechanical ventilation (no longer than 48 h at baseline) with an expected need for continuous invasive ventilation and sedation of at least 24 h and receiving propofol were randomized to propofol or isoflurane sedation for up to 48 ± 6 h. Full inclusion and exclusion criteria of the underlying randomized controlled trial are presented in Appendix A.

The ACD-L was used for isoflurane-treated patients during the first half of the underlying trial. After approval of the ACD-S by the regulatory authorities, an amendment to the study protocol was filed (protocol version 4.0 final, 24 June 2018) that stipulated use of the new device for all patients with expected tidal volumes below 800 mL throughout the rest of the underlying trial. All data were digitally extracted from the trial’s data management system for the first 24 h after study inclusion.

### 2.2. Drug Administration

Isoflurane (Isoflurane 100%, Piramal Pharma, Mumbai, India) was administered via the Sedaconda Anaesthetic Conserving Device (Sedaconda ACD, Sedana Medical AB, Danderyd, Sweden) as recommended by the manufacturer. Propofol 20 mg⋅ml^−1^ (Propofol Hexal, Hexal AG/Sandoz, Holzkirchen, Germany) was infused via a syringe pump. Both drugs were titrated to reach sedation depths of −1 to −4 on the Richmond Agitation Sedation Scale (RASS).

### 2.3. Mechanical Ventilation

All patients were invasively ventilated via oral endotracheal tubes or tracheal cannulas. Either fully controlled mechanical ventilation or pressure-supported spontaneous ventilation modes were used. Ventilation parameters were set according to local protocols and at the discretion of the attending treatment teams. Patients sedated with propofol were ventilated with conventional HMEs (internal volume: 35–50 mL). The two largest centers of the underlying trial (Homburg and Bochum) that included 45% of the patients used HMEs with an internal volume of 35 mL. None of the trial’s centers used active humidification of inspiratory gases.

### 2.4. Measurements

Variables of ventilation, sedation, and analgesia were recorded at 4 h intervals. Results from blood gas analyses closest to the 4 h observation time points were included. The simplified acute physiology score II (SAPS II) was calculated according to Le Gall et al. [15]. Ideal body weight was calculated according to the sex-specific ARDSnet formulas. To assess differences in volatile anesthetic consumption of the two volatile anesthetic conserving devices independently of minute ventilation, isoflurane dose rate was normalized to minute ventilation.

### 2.5. Outcomes

The primary outcome was minute ventilation under controlled and spontaneous ventilation. Secondary outcomes were the following ventilation parameters: tidal volume, respiratory rate, set inspiratory pressure or pressure support, arterial carbon dioxide partial pressure, and volatile anesthetic consumption.

### 2.6. Statistical Analysis

Statistical analyses were carried out with SAS ^®^ version 9.4 (SAS institute, Cary, NC, USA). Normality of numerical baseline characteristics was assessed by visual assessment of histograms, quantile-quantile plots, and Shapiro–Wilk testing. Depending on data distribution, we present continuous measures as mean or median values with the corresponding standard deviation or interquartile ranges for descriptive data. Categorical variables are presented as frequencies (percentages).

Baseline balance is presented as the maximum absolute standardized difference observed between any two groups, where absolute standardized difference is defined as the absolute difference in mean values divided by the pooled standard deviation.

Ventilation parameters were assessed every 4th hour during the randomized study sedation. For the analysis, values were assumed to represent the patients’ parameters for the following 4 h until next assessment. Assigning ventilation parameters to describe 4 h periods, 0–4 h, 4–8 h, and so on, enabled derivation of further analysis variables, combining the above with dosing information which was also derived by 4 h interval. Because of the varying sedation durations, this substudy restricted the comparison between groups to the first 24 h. To further aid evaluability, the 0–4 h period, which presented a low number of available assessments, was excluded.

The differences between treatment groups over time, during the first 24 h, in minute ventilation (L/min), tidal volume (mL), respiratory rate (breaths/min), and arterial carbon dioxide partial pressure (mmHg) were analyzed using linear mixed-effects repeated-measures models with treatment group, time, and time × treatment group interaction as fixed effects and center/pseudo-center as a random effect. Compound symmetry covariance matrices were used to model the within-subject error. Average least-square mean values and differences over the five repeated measurements are presented. The underlying trial was randomized by center. Thus, center/pseudo-center effects were included in the analysis. Pseudo-centers were established to avoid small strata in the analyses. This was managed by merging sites with less than ten patients into pseudo-centers by site type. The site types were defined prior to database lock. Site types in the study were neurological, surgical, internal medicine, general, or other intensive care units. A subgroup analysis was performed for patients having chronic obstructive pulmonary disease (COPD).

To compare effects of the volatile anesthetic application device on the relationship of exhaled isoflurane vapor volume per breath (mL) and isoflurane consumption per minute ventilation ((mL/h)/(L/min)), we used linear regression. Using the data from the regression analysis, isoflurane consumption with specified ventilation parameters, isoflurane concentration, and the respective reflector can be calculated as follows (abbreviations are outlined in Table 1):IR/MV = slope × VVIR/MV = slope × VT × c-Iso × (100 Vol%)^−1^IR = slope × VT × c-Iso × (100 Vol%)^−1^ × RR × VT × (1000 mL/L)^−1^(1)

To illustrate the impact of the respective reflector on isoflurane consumption, with a special focus on the necessary increase in minute ventilation to compensate the higher dead space of ACD-L, a three-dimensional graph was constructed with the x-axis representing respiratory rate, the y-axis representing tidal volume, and the z-axis representing the remaining factors of the right side of the above equation:z-axis: slope × VT × c-Iso × (100 Vol%)^−1^ × (1000 mL/L)^−1^(2)

The volume of the resulting blocks will thus visualize the consumption of isoflurane under the specified circumstances.

## 3. Results

### 3.1. Study Population Characteristics

The underlying trial included a total of 301 patients: 150 were randomized to isoflurane sedation and 151 were randomized to propofol sedation. Moreover, 64 patients received isoflurane via the ACD-S and 86 patients via the ACD-L. One patient that received isoflurane via the ACD-S was excluded from all analyses due to incomplete data, leaving a total of 63 ACD-S patients for the current study. Some patients were excluded due to incomplete data only for specific analyses (Figure 1).

Most baseline patient characteristics were well balanced among treatment groups, while the number of patients suffering from chronic obstructive pulmonary disease (COPD) was slightly higher in the ACD-L than in the propofol and ACD-S treatment groups (Table 2). The overall fraction of patients breathing spontaneously under pressure-supported ventilation for more than 10% of the observation time was low, with 19% (28/151) for the propofol, 40% (25/63) for the ACD-S, and 29% (25/86) for the ACD-L treatment groups.

### 3.2. Primary Outcome—Minute Ventilation

Average minute ventilation was approximately 1 L/min higher during isoflurane sedation via the ACD-L device compared to the ACD-S device or propofol sedation (average difference (95%CI) ACD-L vs. ACD-S: 1.3 (0.6, 2.0) L/min, *p* < 0.001; ACD-L vs. Propofol: 1.3 (0.7, 1.8) L/min, *p* < 0.001; Figure 2). In contrast, average minute ventilation did not differ between propofol sedation and isoflurane sedation via the ACD-S (−0.1 (−0.7, 0.6) L/min, *p* = 0.823; Figure 2).

### 3.3. Ventilation Parameters

Ventilation parameters are presented in Figure 2 with average differences and the corresponding 95% confidence intervals (95%CI). No time-related changes in ventilation parameters and arterial carbon dioxide partial pressures were observed (Appendix A). Tidal volumes were significantly higher during isoflurane sedation via the ACD-L compared to propofol sedation (average difference (95%CI): 44 (16, 72) mL, *p* = 0.002; Figure 2), while tidal volumes were only slightly higher during ACD-S usage without reaching statistical significance (ACD-S vs. propofol: 24 (−7, 55) mL, *p* = 0.126; Figure 2). Respiratory rate and arterial carbon dioxide pressure were again significantly higher during isoflurane sedation via the ACD-L, but no significant differences were observed between propofol sedation and isoflurane sedation via the ACD-S (Figure 2; Appendix A). Set inspiratory pressure and inspiratory pressure support did not differ among treatment groups (Appendix A).

The effect of the ACD-L on minute ventilation was more prominent in COPD patients. The average difference in minute ventilation in COPD patients was 2.8 (0.7, 4.9) L/min (*p* = 0.011) between ACD-L and ACD-S, and 2.0 (0.3, 3.7) L/min, (*p* = 0.024) between ACD-L and propofol sedation; there was no statistically significant difference between ACD-S and propofol sedation: −0.8 (−3.2, 1.6) L/min (*p* = 0.511). Neither tidal volume, respiratory rate, nor arterial carbon dioxide pressures in COPD patients differed among treatment groups (Appendix A).

### 3.4. Volatile Anesthetic Consumption and Reflection Efficiency

End-tidal isoflurane concentrations did not significantly differ between the ACD-S and ACD-L devices (average difference (95% confidence interval (95%CI)): 0.04 (−0.03, 0.10) %, *p* = 0.287; Table 3; Appendix A). Isoflurane pump rates were significantly lower when ACD-L was used compared to ACD-S (average difference (95%CI): −0.7 (−1.3, 0.1) mL/h, *p* = 0.022; Table 3, Appendix A). Normalization of isoflurane dose rate to minute ventilation also revealed significantly lower volatile anesthetic consumption with the ACD-L than with the ACD-S device (average difference (95%CI): −0.13 (−0.20, −0.07) (mL/h)/(L/min), *p* < 0.001; Appendix A).

Figure 3 shows the effect of ACD-S and ACD-L on the relationship between the isoflurane dose rate normalized to minute ventilation and the exhaled isoflurane vapor volume in one breath. Linear regression lines give reasonably good fits. The slope of the regression line for ACD-L is 30% lower than that for ACD-S, indicating a higher reflection efficiency.

Using these estimates, the isoflurane consumption can be calculated as the required pump rate under various conditions (slope times minute volume times vapor volume per breath, Figure 4). Without any compensatory increase in minute ventilation, 30% of the anesthetic would be saved when using the ACD-L (Figure 4A,B). However, under clinical conditions, ventilatory compensations for the larger dead space of the ACD-L partially reverse these savings in anesthetic. The calculations in Figure 4 use standard parameters of ventilation, based on the mean values of the patients of this study, and on published data on the ventilatory dead space effect of the two medical devices. To be precise, in addition to the volumetric dead space, which is equivalent to the internal volume (50 mL in ACD-S versus 100 mL in ACD-L), the reflective dead space (25 mL in ACD-S versus 40 mL in ACD-L) due to carbon dioxide reflection must also be considered [12]. In total, this results in an apparent device dead space of 75 mL for the ACD-S and 140 mL for the ACD-L. Thus, a compensation of the higher dead space ventilation with the ACD-L via respiratory rate reduces savings to 17% (Figure 4C), and compensation via tidal volume reduces savings to 10% (Figure 4D).

## 4. Discussion

Isoflurane sedation via the smaller ACD-S reduced minute ventilation by 1.3 L/min on average compared to the ACD-L. Use of the ACD-L increased tidal volume by around 40–50 mL and inspiratory rate by 1–2 breaths/min, while use of the ACD-S did not significantly affect ventilation compared to a standard mechanical ventilation and sedation setting. One may, therefore, exclude clinically important increases in ventilatory demands under ACD-S usage. We note, however, that minimally higher tidal volumes under usage of the ACD-S are physically plausible and our data suggest compensations in tidal volume by about 20 mL, although these were not statistically significant.

Our findings are in line with previous smaller crossover studies that alternately tested the ventilatory effects of using the ACD-S or ACD-L device [13,16]. The first study included ten spontaneously breathing intensive care patients and reported almost identical effects on ventilation, with average increases in minute ventilation by 1 L/min and tidal volume by 66 mL during ACD-L use [13]. Another study found considerable increases in arterial carbon dioxide without significant differences in ventilation parameters; however, all patients underwent volume-controlled ventilation strictly adapted to body weight not allowing compensatory increases in tidal volume [16]. Taken together, use of the ACD-L impedes carbon dioxide elimination or in turn causes compensatory increases in minute ventilation, whereas use of the ACD-S leaves ventilation parameters and arterial carbon dioxide pressures largely unaffected.

The total ventilatory dead space of anesthetic conserving devices consists of volumetric and reflective dead space [17,18,19]. While the volumetric dead space represents the device’s internal volume, reflective dead space arises from carbon dioxide reflection by the activated charcoal membrane of the anesthetic conserving device. Initial laboratory studies suggested considerable carbon dioxide reflection [17]. However, under consideration of clinical conditions (presence of humidity and volatile anesthetics), only marginal carbon dioxide reflection with reflective dead spaces of 25 mL for the ACD-S of and 40 mL for the ACD-L were reported [12]. Therefore, total ventilatory dead spaces are 75 mL for the ACD-S and 140 mL for the ACD-L, compared to a ventilation with conventional heat and moisture exchangers with a dead space of 35 mL under propofol sedation [12]. Whereas the ACD-L considerably affected ventilation, our data suggest that the dead space of the ACD-S is compensated by slightly higher tidal volumes (about 20 mL), although they are not statistically significant.

Inhaled sedation may offer several benefits for the treatment of patients suffering from acute respiratory distress syndrome (ARDS) including decreased airway resistance [20], adequate sedation depths needed for prone positioning or continuous lateral rotation [21,22,23], better maintenance of spontaneous breathing [24,25], anti-inflammatory properties [26,27], less epithelial lung injury, and improved oxygenation [28]. On the other hand, use of anesthetic conserving devices in ARDS patients was previously questioned for low tidal volume ventilation due to the larger ventilatory dead space causing higher ventilatory demands [11]. However, these concerns were based on the older ACD-L device with larger dead space. As we did not detect substantial influences of isoflurane sedation via the ACD-S compared to conventional propofol sedation and use of HMEs on ventilation parameters, we conclude that clinicians should not refrain from using the ACD-S for isoflurane sedation in ARDS patients, at least in centers that use HMEs as standard of care.

Volatile anesthetic consumption of anesthetic conserving devices depends on three major factors—the target anesthetic concentration, minute ventilation, and reflection efficiency [29]. Therefore, the smaller size of the ACD-S results not only in lower ventilatory demands but also in a lower reflection efficiency. Consistently, mean isoflurane consumption was 0.7 mL/h (18%) lower in patients receiving isoflurane via the ACD-L, although end-tidal isoflurane concentrations were similar with both devices. Our calculations show that savings with the ACD-L could be as high as 30%. However, considering compensatory increases in minute ventilation to account for the additional dead space, savings may only range between 10 and 17%. The observed 18% lower isoflurane consumption with the ACD-L can be explained by the fact that additional dead space was not fully compensated, as evident from higher arterial carbon dioxide pressures under ACD-L usage. Consequently, higher demands in minute ventilation with the ACD-L largely outweigh its higher reflection efficiency.

We note, however, that most of the patients included in our study had end-tidal isoflurane concentrations below 0.6% and tidal volumes below 800 mL, meaning that the exhaled isoflurane vapor volume and minute ventilation did not exceed the reflection capacity of the ACD-S [14]. We assume that the difference in volatile anesthetic consumption will become more apparent within higher ranges of tidal volumes [14]. As previously shown in L-type COVID-19 patients (high compliance, high tidal volumes, and high minute ventilation), volatile anesthetic reflection with the ACD-S may gradually become insufficient when tidal volumes exceed 800 mL and the less potent drug sevoflurane is used [23]. In addition, since the relative contribution to ventilatory dead space is less in patients with high tidal volumes, carbon dioxide rebreathing is less of a problem. Thus, the ACD-L version may still be a preferred option for rare cases when patients generate high minute ventilation and require high doses of volatile anesthetics.

Our data suggest that the larger ACD-L should be used with caution in COPD patients due to noticeable increases in minute ventilation. As neither tidal volume nor respiratory rate differed among treatment groups in COPD patients, increases in overall minute ventilation cannot be attributed to a common pattern of ventilatory dead space compensation. Similar arterial carbon dioxide pressures across groups suggest that most of the increase in dead space was sufficiently compensated by increased minute ventilation. In contrast, use of the ACD-S did not interfere with ventilation in COPD patients when compared to a standard mechanical ventilation and sedation setting, and should thus be the device of choice when inhaled sedation is desired in these patients. We note though that the subgroup of COPD patients was small, and we have no information on disease stages, limiting generalizability of these findings.

This study has limitations. Patients were randomized to isoflurane and propofol sedation but not to the volatile anesthetic reflection devices. However, after changing the applied device, the study was continued under the same conditions, making a historical bias unlikely. We have no information on the size of the HMEs used at smaller centers, but the two largest centers contributing 45% of the patients used HMEs with an internal volume of 35 mL.

## 5. Conclusions

Isoflurane sedation via the Sedaconda ACD-L compared to the ACD-S or propofol sedation leads to increases in carbon dioxide levels and compensatory increases in minute ventilation. In contrast, isoflurane sedation via the ACD-S has no substantial influence on ventilation compared to a standard mechanical ventilation and sedation setting. The smaller ACD-S is therefore the device of choice to minimize impact on ventilation, especially in patients with a limited ability to compensate (e.g., COPD patients). Volatile anesthetic consumption is slightly higher with the smaller ACD-S but the difference largely disappears under clinical conditions for tidal volumes below 800 mL.

## Figures and Tables

**Figure 1 jcm-12-03314-f001:**
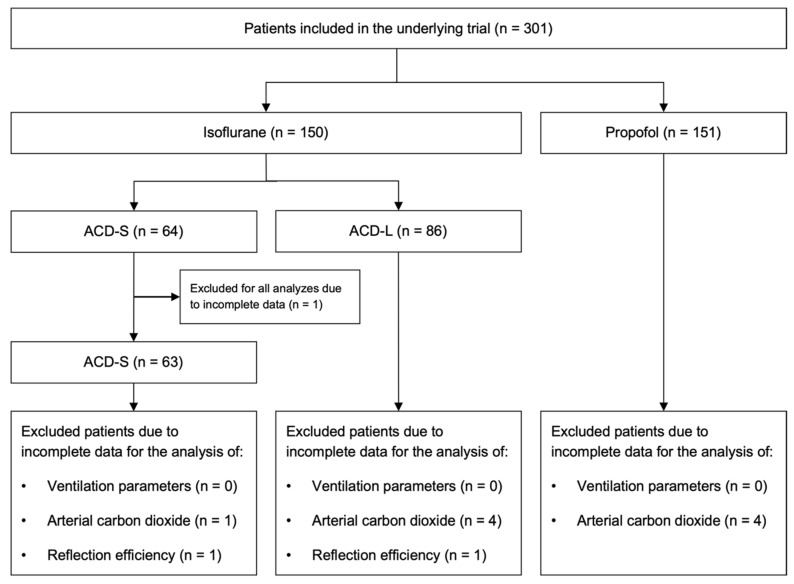
Patient flow chart.

**Figure 2 jcm-12-03314-f002:**
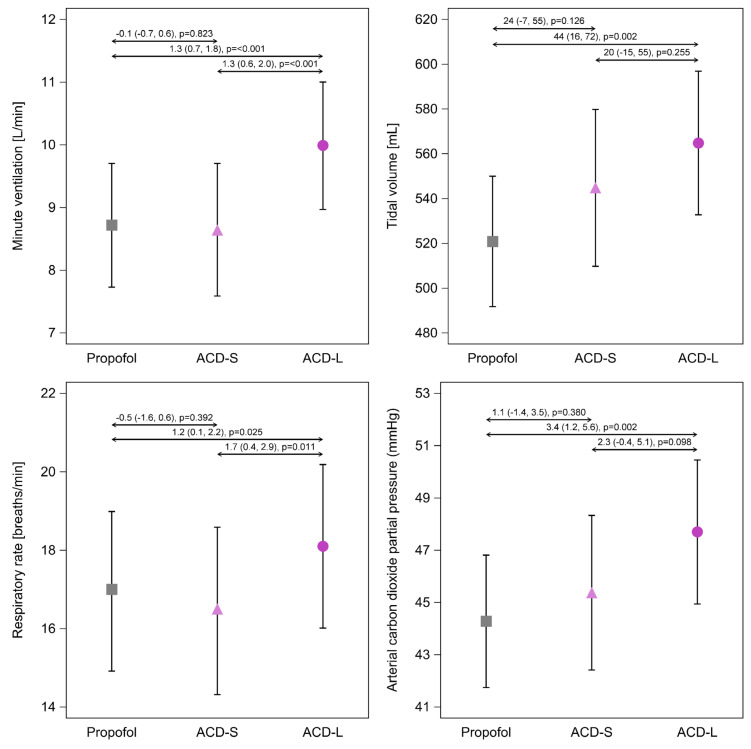
Comparison of ventilation parameters. Data are presented as mean values and 95% confidence intervals (95%CI). Effect sizes are presented as average differences (95%CI) calculated by linear mixed-effects models. ACD-S, anesthetic conserving device with 50 mL internal volume. ACD-L, anesthetic conserving device with 100 mL internal volume.

**Figure 3 jcm-12-03314-f003:**
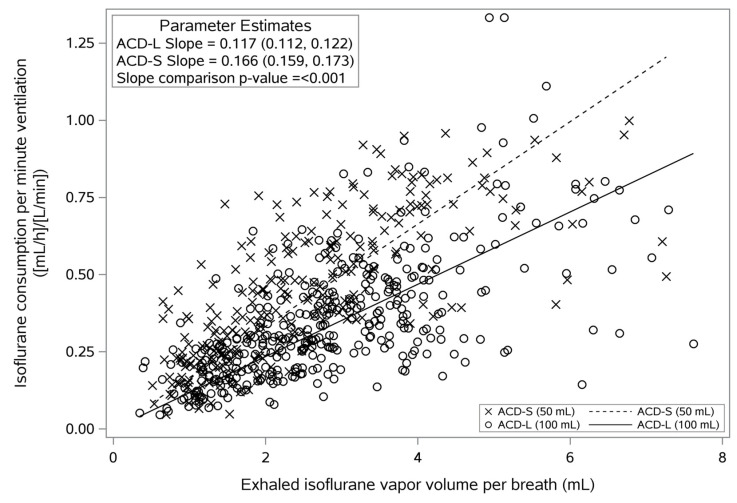
Comparison of volatile anesthetic reflection efficiency. Dots represent raw data. Linear regression with the intercept set to zero was used to visualize the differences in reflection efficiency between ACD-S and ACD-L over the range of exhaled isoflurane vapor volumes. ACD-S, anesthetic conserving device with 50 mL internal volume. ACD-L, anesthetic conserving device with 100 mL internal volume.

**Figure 4 jcm-12-03314-f004:**
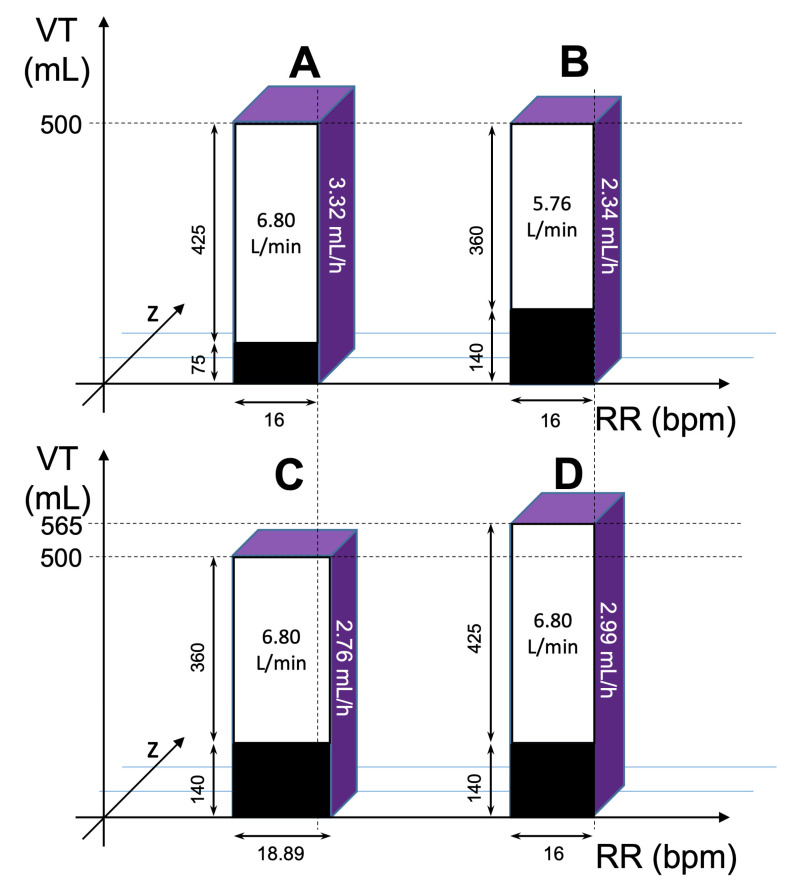
Effects of ventilation on isoflurane consumption of ACD-S and ACD-L. The volume of each block represents the isoflurane consumption (numerical value printed in white on the right side). For its calculation, the slopes of the regression lines from Figure 3, a tidal volume of 500 mL (tidal volume (VT), height of each block), a respiratory rate of 16 bpm (respiratory rate (RR), width of each block), and an isoflurane concentration of 0.5 Vol% (leading to 2.5 mL isoflurane vapor exhaled in one breath of 500 mL) were assumed. Total device dead space: 75 mL for ACD-S and 140 for ACD-L. z-axis with arbitrary units (for a detailed calculation of the z-axis, see text). Black areas on the front sides of the blocks represent dead space ventilation, white areas the remaining alveolar ventilation. (**A**) ACD-S; (**B**) ACD-L with equal minute ventilation; (**C**) ACD-L compensation of increased dead space ventilation by higher respiratory rate; (**D**) ACD-L compensation by a higher tidal volume. Note that with the higher tidal volume, reflection efficiency decreases, and the depth of block D is greater.

**Table 1 jcm-12-03314-t001:** Abbreviations of the isoflurane consumption formula.

Variable	Unit	Description
IR	(mL/h)	Isoflurane pump rate (in steady state this will reflect isoflurane consumption)
VT	(mL)	Tidal volume
RR	(min^−1^)	Respiratory rate
MV	(L/min)	Minute ventilation (MV can be calculated by multiplying RR with VT × (1000 mL/L)^−1^)
Slope	(mL^−1^)	Slope of the regression line (0.166 Vol%^−1^ × mL^−1^ for ACD-S or 0.117 Vol%^−1^ × mL^−1^ for ACD-L)
c-Iso	(Vol%)	End-tidal isoflurane concentration
VV	(mL)	Isoflurane vapor volume contained in one breath (VV is calculated by multiplying VT and c-Iso × (100 Vol%)^−1^)

**Table 2 jcm-12-03314-t002:** Study population characteristics.

Parameter	Propofol	ACD-S	ACD-L	SMD
*n*	151	63	86	-
Sex (male)	98 (65)	41 (65)	62 (72)	0.16
Age (years)	64 ± 13	66 ± 12	66 ± 12	0.12
Height (cm)	173 ± 9	172 ± 9	174 ± 8	0.08
Weight (kg)	84 ± 23	84 ± 21	84 ± 17	0.01
BMI (kg/m^2^)	28 ± 8	29 ± 7	28 ± 5	0.03
SAPS II	41 ± 18	38.3 ± 17	40 ± 17	0.17
COPD (*n*)	14 (9)	7 (11)	17 (20)	0.30

Data are reported as mean ± standard deviation or numbers (percentages). The maximum standardized mean difference (SMD) observed between two treatment groups is presented as a measure of balance. ACD-S, anesthetic conserving device with 50 mL internal volume. ACD-L, anesthetic conserving device with 100 mL internal volume. BMI, body mass index. SAPS II, simplified acute physiology score II. COPD, chronic obstructive pulmonary disease.

**Table 3 jcm-12-03314-t003:** Sedative drugs.

Variable	Propofol	ACD-S	ACD-L	Average Difference	*p*
End-tidal isoflurane concentration [%]	-	*n = 64* 0.44 (0.40, 0.49)	*n = 85* 0.46 (0.42, 0.50)	0.04 (−0.03, 0.10)	0.287
Isoflurane pump rate [mL/h]	-	*n = 64* 3.7 (2.9, 4.5)	*n = 86* 3.0 (2.3, 3.8)	−0.7 (−1.3, 0.1)	0.022
Propofol infusion rate [mg/kg/h]	*n = 149* 2.4 (2.2, 2.6)	-	-	-	-

Data are presented as mean values and 95% confidence intervals (95%CI). Effect sizes are presented as average differences (95%CI) calculated by linear mixed-effects models. ACD-S, anesthetic conserving device with 50 mL internal volume. ACD-L, anesthetic conserving device with 100 mL internal volume.

## Data Availability

Data are available from the authors upon reasonable request.

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
