# Peer review of "Ventilatory Effects of Isoflurane Sedation via the Sedaconda ACD-S versus ACD-L: A Substudy of a Randomized Trial"

_jcm, 2023, doi:10.3390/jcm12093314_

Round 1

Reviewer 1 Report

Dear Authors,

Main comments:

pg 5 of 14, lines 197-198:This is not a light bias. This because the COPD pts have a reduced capability of CO2 elimination and a Vdphys increased, so the impact of a larger artificial Vd is not negligible. 

pg 10 of 14, lines 313-314: in patients with normal respiratory mechanics the higher Vdtot is easily compensated by the increase of MV, and it don't expose the patient to difficult ventilation setting or dangerous ventilaory approach. Differently, in COPD pts with a reduce clearance of CO2, this could be an important evidence. I would like to see the results of the subgroup of COPD  pts, and if it is to address the difference.

fig 2: Could you describe the COPD pts separately?

Minor:

pg 2 of 14: "as commonly" it is not so commun and widespread

pg 4 of 14, lines 159-173: this list it could more clear if edited as a table

Author Response

Dear Authors,

Main comments:

pg 5 of 14, lines 197-198:This is not a light bias. This because the COPD pts have a reduced capability of CO2 elimination and a Vdphys increased, so the impact of a larger artificial Vd is not negligible.

pg 10 of 14, lines 313-314: in patients with normal respiratory mechanics the higher Vdtot is easily compensated by the increase of MV, and it don't expose the patient to difficult ventilation setting or dangerous ventilatory approach. Differently, in COPD pts with a reduced clearance of CO2, this could be important evidence. I would like to see the results of the subgroup of COPD  pts, and if it is to address the difference.

fig 2: Could you describe the COPD pts separately?

Authors: Thank you very much for these important remarks. Based on your comments, we performed a subgroup analysis for patients diagnosed with COPD. We found that the effect of the ACD-L on minute ventilation is more prominent in COPD patients, while there was no significant effect on minute ventilation when the ACD-S was used. These are interesting new findings, and we much appreciated the reviewer's input leading us to the extension of our investigation. Our new findings are presented in the results section and discussed in the discussion sections, as given below. A detailed presentation of the results of the subgroup analysis was added to the supplement.

Results: “The effect of the ACD-L on minute ventilation was more prominent in COPD patients. The average difference in minute ventilation in COPD patients was 2.8 [0.7, 4.9] L/min (p=0.011) between ACD-L and ACD-S, and 2.0 [0.3, 3.7] L/min, (p=0.024) between ACD-L and propofol sedation; there was no statistically significant difference between ACD-S and propofol sedation: -0.8 [-3.2, 1.6] L/min (p=0.511). Neither tidal volume, respiratory rate, nor arterial carbon dioxide pressures in COPD patients differed among treatment groups (Supplement, Figures S11-S14).”

Discussion: “Our data suggest that the larger ACD-L should be used with caution in COPD patients due to noticeable increases in minute ventilation. As neither tidal volume nor respiratory rate differed among treatment groups in COPD patients, increases in overall minute ventilation cannot be attributed to a common pattern of ventilatory dead space compensation. Similar arterial carbon dioxide pressures across groups suggest that most of the increase in dead space was sufficiently compensated by increased minute ventilation. In contrast, use of the ACD-S did not interfere with ventilation in COPD patients when compared to a standard mechanical ventilation and sedation setting, and should thus be the device of choice when inhaled sedation is desired in these patients. We note though that the subgroup of COPD patients was small, and we have no information on disease stages, limiting generalizability of these findings.”

Minor:

pg 2 of 14: "as commonly" it is not so common and widespread.

Authors: We apologize for the ambiguous meaning of the sentence. “as commonly” was supposed to refer to the open circuit ventilators but not inhaled sedation. We therefore revised the sentence as follows: “Volatile anesthetic reflection devices were designed to enable inhaled sedation with open circuit ventilators which are commonly used in intensive care.“

pg 4 of 14, lines 159-173: this list, it could be clearer if edited as a table.

Authors: We have edited the list as a table to enhance clarity.

Reviewer 2 Report

In their paper "Ventilatory effects of isoflurane sedation via the Sedaconda ACD-S versus ACD-L: a substudy of a randomized trial" the authors give information about the impact on ventilatory parameters by using two devices for inhaled sedation of the same manufacturer differing in death space. Both devices (in fact the same device, but different size/death space) are compared to i.v. sedation and ventilation via a HME.

Brief information about the number, size, tape an location of the study centers might be informative. 

Different type of spelling is used for the word An(a)esthetic throughout the article.

Figure 3 might be easier to read if different colors could be used.

Should the employment by Sedana Medical AB be stated as a affiliation?

Author Response

In their paper "Ventilatory effects of isoflurane sedation via the Sedaconda ACD-S versus ACD-L: a substudy of a randomized trial" the authors give information about the impact on ventilatory parameters by using two devices for inhaled sedation of the same manufacturer differing in death space. Both devices (in fact the same device, but different size/death space) are compared to i.v. sedation and ventilation via a HME.

Authors: We are grateful for your constructive comments that have helped us considerably to improve our manuscript.

Brief information about the number, size, tape and location of the study centers might be informative.

Authors: We have added the above-suggested information to the manuscript methods section, as follows: “The underlying trial investigated the non-inferiority of isoflurane to propofol sedation in intensive care patients, and included patients from 13 medical or general, 10 surgical, and 1 neurological intensive care units in Germany (21 sites) and Slovenia (3 sites).”

Different type of spelling is used for the word An(a)esthetic throughout the article.

Authors: The overall manuscript is written in American English. However, the “Anaesthetic Conserving Device” is the manufacturer-given name that includes the British spelling. For reasons of consistency with previous publications on the topic, we chose to keep the previous spelling.

Figure 3 might be easier to read if different colors could be used.

Authors: To enhance the readability of figure 3, we switched to black circles and crosses.

Should the employment by Sedana Medical AB be stated as an affiliation?

Authors: The employment of Peter Sackey by Sedana is now stated as an additional affiliation.

Round 2

Reviewer 1 Report

Thanks for addressing the suggestions. No more comments.